# Contribution of Different Food Types to Vitamin A Intake in the Chinese Diet

**DOI:** 10.3390/nu15184028

**Published:** 2023-09-17

**Authors:** Xue Li, Can Guo, Yu Zhang, Li Yu, Fei Ma, Xuefang Wang, Liangxiao Zhang, Peiwu Li

**Affiliations:** 1Key Laboratory of Biology and Genetic Improvement of Oil Crops, Ministry of Agriculture and Rural Affairs, Oil Crops Research Institute, Chinese Academy of Agricultural Sciences, Wuhan 430062, Chinamafei01@caas.cn (F.M.);; 2Quality Inspection and Test Center for Oilseed Products, Ministry of Agriculture and Rural Affairs, Oil Crops Research Institute, Chinese Academy of Agricultural Sciences, Wuhan 430062, China; 3College of Food Science and Engineering, Collaborative Innovation Center for Modern Grain Circulation and Safety, Nanjing University of Finance and Economics, Nanjing 210023, China; 4Hubei Hongshan Laboratory, Wuhan 430070, China; 5Xianghu Laboratory, Hangzhou 311231, China

**Keywords:** vitamin A, vegetables, dietary intake, vegetable oil fortification

## Abstract

Vitamin A is a fat-soluble micronutrient that is essential for human health. In this study, the daily vitamin A intake of Chinese residents was evaluated by investigating the vitamin A content of various foods. The results show that the dietary intake of vitamin A in common foods was 460.56 ugRAE/day, which is significantly lower than the recommended dietary reference intake of vitamin A (800 ugRAE/day for adult men and 700 ugRAE/day for adult women). Vegetables contributed the most to daily vitamin A dietary intake, accounting for 54.94% of vitamin A intake (253.03 ugRAE/day), followed by eggs, milk, aquatic products, meat, fruit, legumes, coarse cereals, and potatoes. Therefore, an increase in the vitamin A content of vegetables and the fortification of vegetable oils with vitamin A are effective ways to increase vitamin A intake to meet the recommended dietary guidelines in China. The assessment results support the design of fortified foods.

## 1. Introduction

Vitamin A is a compound with the biological activity of retinol. There are two primary groups of substances that provide retinol bioactivity. One is preformed vitamin A (retinol) and the other is provitamin A carotenoids chiefly comprising beta-carotene, alpha-carotene, and cryptoxanthin [1,2]. As an essential fat-soluble micronutrient, vitamin A cannot be synthesized by the human body and needs to be obtained from a regular diet. Vitamin A plays a vital role in ocular function and is involved in preventing xerophthalmia, the maintenance of eye integrity, and cellular division and differentiation [3,4,5]. Recently, the global prevalence of dry eye disease (DED) has been estimated to be between 5% and 50% [6]. It was concluded from previous investigations that excessive smartphone use in children and adolescents is associated with DED [7,8,9]. It was concluded in another study that people who use a visual display terminal (VDT) for over 3.71 h per day and occupational VDT users are susceptible to developing DED [10]. Moreover, vitamin A is associated with reproduction, the immune system, and bone and embryonic development [11,12,13]. Vitamin A deficiency (VAD) has drawn more and more attention from all over the world, and the World Health Organization (WHO) confirmed VAD is one of four major nutritional deficiency diseases. It is common to see chronic VAD and it is hard to detect because it is asymptomatic [14]. Children are commonly found to have VAD, especially preschool-aged children, and boys and girls possess incidences of 0.8% and 0.4%, respectively [15,16,17]. VAD is the primary cause of childhood xerophthalmia and blindness in developing countries [18] and is associated with poor clinical outcomes in patients with sepsis [19]. In addition, 10–20% of pregnant women in the world lack sufficient vitamin A intake. If lactating women do not intake vitamin A, it will affect the concentration of vitamin A in their infants. As a result, infants with VAD will have wheezing symptoms [20,21]. In addition, the newborns of pregnant women with gestational diabetes have an increased impairment of vitamin A formation [22]. Therefore, increasing vitamin A intake is important to protect the eyes and reduce the incidence of VDA.

Generally, retinol from animal-derived foods and provitamin A carotenoids from plant-derived products are the two primary sources of vitamin A. Retinol can be obtained from eggs, animal liver, fish, and dairy products. The main source of provitamin A carotenoids are yellow and orange fruits as well as dark-green vegetables [23]. β-carotene shows a higher provitamin A activity than β-cryptoxanthin and α-carotene [24]. The dietary intake of vitamin A in Western countries is different from that in developing countries. The typical daily diet in Western countries provides an approximate intake of 20–34% of provitamin A carotenoids, while more than 70% of provitamin A carotenoids were ingested from diets in developing countries [25]. Furthermore, vitamin A-fortified products have appeared on the market, such as fortified wheat flour and milk [26,27]. However, whether the body is deficient in vitamin A and whether more vitamin A should be obtained from the diet and fortified products is not clear. Moreover, acute toxicity can occur if an individual’s intake of vitamin A is greater than 3000 ugRAE/day [28,29]. Therefore, it is important to evaluate the vitamin A content in people’s daily diets.

A dietary assessment plays an important role in guiding diet intake. It is an investigation of the relationship between diet and health conditions. The US Institute of Medicine recommends estimating the degree of inadequate dietary nutrient intake in a group according to the Estimated Average Requirements (EARs) [30]. The Recommended Dietary Allowance (RDA) is the average daily dietary intake which is adequate to meet the nutritional needs among healthy people. There are many methods used to assess food intake, including food frequency questionnaires (FFQs), 24 h recalls, and weighed and non-weighed food records [31]. Each method has its own advantages and disadvantages. Among these, weighed food records are the gold-standard recommendation, but they are time-consuming and vary depending on the participant [32]. FFQs are commonly used in large dietary epidemiology studies due to their low cost and time efficiency, but they lack specificity [33]. In addition, novel approaches such as image-assisted and image-based methods are applied with the use of mobile devices [34]. In a previous study, the contribution of different foods to the total phytosterol intake of Chinese residents was estimated according to consumption data [35]. In another work, the content of vitamin E in different foods was used to assess vitamin E intake in individuals’ daily diet [36]. The simplified dietary assessment (SDA) method was proposed by the International Vitamin A Consultative Group (IVACG) to identify and monitor groups at risk for suboptimal vitamin A intake [37]. The overweight and obese Dominican population was investigated to assess vitamin A and carotenoid intake [38]. However, there is no study that reported on the dietary vitamin A intake in the Chinese diet. Therefore, conducting a dietary assessment of the Chinese population is essential.

In this study, the vitamin A contents in different types of foods and the amount of vitamin A intake in the Chinese daily diet are investigated. In addition, strategies are proposed to increase the vitamin A intake in China.

## 2. Materials and Methods

### 2.1. Data Sources

Cereals, vegetable oils, nuts, coarse cereals, potatoes, legumes, vegetables, fruits, meat, eggs, milk, and aquatic products are used to obtain vitamin A according to the Chinese Food Composition Tables released by the National Institute for Nutrition and Health and Chinese Center for Disease Control and Presentation; the production, supply, and distribution (PSD) reports released by the United States Department of Agriculture (USDA) and the Food and Agriculture Organization of the United States (FAOSTAT); the China Population Nutrition and Health Status Monitoring Report; and the China Statistical Yearbook providing the domestic consumption of the main kinds of foods in China. The consumption of major foods in the Chinese diet, including cereals, coarse cereals, potatoes, legumes, vegetables, fruits, nuts, vegetable oils, meat, eggs, milk, and aquatic products, was determined according to the China Population Nutrition and Health Status Monitoring Report and USDA PSD reports. The 16 most frequently consumed kinds of vegetables were selected, and the consumption of each kind of vegetable was obtained from FAOSTAT, while the USDA PSD reports provided the most consumed fruits, which included apples, bananas, pears, grapes, peaches, tangerines, oranges, grapefruits, and cherries. In summary, 45 kinds of foods were used to evaluate vitamin A intake in the Chinese diet.

### 2.2. Calculation Method

The United States Health and Medicine Division (HMD) defined total vitamin A intake as the sum of 1 μg of retinol, 1/12 μg of dietary β-carotene, and 1/24 μg of dietary α-carotene, expressed as μg of retinol activity equivalents (RAEs). First, the estimated daily vitamin A intake was calculated according to total vitamin A content and the consumption of various foods that are commonly consumed; meanwhile, the contribution of one kind of food to the total vitamin A intake was calculated. Second, the amount of each vegetable required to reach the recommended vitamin A intake if a person were to eat only 1 of the 16 types of vegetables was calculated. Similarly, the amount of each vegetable required if the daily intake of the other 15 vegetables remained unchanged was calculated. In addition, the increase in vitamin A content in each vegetable needed if the daily intake of the other 15 vegetables remained unchanged was calculated.

The data analysis was carried out in Microsoft Excel Version 2021 (Microsoft Corporation, Redmond, WA, USA).

## 3. Results

### 3.1. Vitamin A Contents in Various Foods

Vitamin A is a fat-soluble vitamin that is essential for the maintenance of human metabolism and is available from various foods in two forms. One is pre-formed vitamin A (retinol) which is found only in animals, the other is provitamin A carotenoids which is found in plants. Cereals are the staple food for the Chinese population, and are rich in nutrients but contain a low content of provitamin A carotenoids. Vegetable oils and nuts are also widely consumed foods, but also contain low content of provitamin A carotenoids. Therefore, the provitamin A carotenoids content of coarse cereals, potatoes, legumes, vegetables and fruits, the retinol content of meat, eggs, milk, and aquatic products were collected and listed in Appendix A. It is evident that vegetables contained the highest content (1088 ugRAE/100 g) of total vitamin A. For example, carrots contained 342 ugRAE/100 g and spinach contained 243 ugRAE/100 g of total vitamin A content, indicating that carrots and spinach are good sources of provitamin vitamin A carotenoids. Meanwhile, eggs have the second highest content of total vitamin A (1045 ugRAE/100 g). All four types of eggs contained high levels of retinol, particularly quail eggs with 337 ugRAE/100 g. In addition, the following retinol contents were detected in aquatic products, milk, and meat. Fruits, beans, potatoes, and coarse cereals had low levels of carotenoids in contrast with other commonly consumed foods.

### 3.2. Evaluation of the Vitamin A Dietary Intake in the Chinese Diet

The nine food categories including coarse cereals, potatoes, legumes, vegetables, fruits, meat, eggs, milk, and aquatic products were used to calculate the dietary intake of vitamin A. As shown in Figure 1, the dietary intake of vitamin A in these nine kinds of commonly consumed foods was 460.56 ugRAE/day, which was consistent with the vitamin A supply in Asia (431 ugRAE/day) estimated by WHO. However, the recommended dietary reference intakes of vitamin A for adult men and women were 800 ugRAE/day and 700 ugRAE/day in China, respectively [39]. As a result, the actual dietary intake of vitamin A remained inadequate for an adult. The contribution of different foods to vitamin A dietary intake is shown in Figure 1. Vegetables are the first contributor to the dietary intake of vitamin A among the nine food categories, accounting for 54.94% of vitamin A intake (253.03 ugRAE/day), followed by eggs with 18.30% (84.30 ugRAE/day), and the others accounting for less than 10%. The animal foods including milk, aquatic products, and meat contributed 9.80% (45.12 ugRAE/day), 9.45% (43.51 ugRAE/day), and 6.15% (28.32 ugRAE/day), respectively; while fruit, legumes, coarse cereals, and potatoes contributed 0.57% (2.64 ugRAE/day), 0.41% (1.87 ugRAE/day), 0.24% (1.12 ugRAE/day), and 0.14% (0.64 ugRAE/day), respectively.

Vegetables were the primary contributor to vitamin A intake, and Table 1 shows the vitamin A content of 16 kinds of vegetables and their contribution to total vitamin A intake. The vitamin A intake of 253.03 ugRAE/day was recorded from vegetables. Carrots and dark green vegetables are rich in provitamin A carotenoids, and the result showed that carrots had the highest contribution to vitamin A intake at 103.49 ugRAE/day and accounted for 28.33% of the 45 kinds of foods. Spinach ranked second at 14.51% (66.83 ugRAE/day). The contributions of Chinese chives, pepper, tomato, asparagus lettuce, lettuce, and pumpkin were 4.78% (22.01 ugRAE/day); 4.13% (19.00 ugRAE/day); 2.62% (12.06 ugRAE/day); 2.57% (11.83 ugRAE/day); 1.96% (9.03 ugRAE/day); and 1.22% (3.08 ugRAE/day), respectively. The total contribution of other vegetables was 1.92% (8.80 ugRAE/day).

## 4. Discussion

The actual average vitamin A daily intake cannot meet the recommended dietary reference intake in the Chinese diet. Nearly 254 million of preschool-aged children were affected by VAD and suffered from night blindness [40]. Meanwhile, the excessive smartphone use potentially increased the DED. The increase in eye diseases requires a greater demand for vitamin A.

Bioavailability was defined as the fraction of an ingested bioactive agent, which reaches the specific site of action in the body. Bioavailability was primarily determined by three factors: bioaccessibility, transformation, and absorption [41]. The main factors limiting the bioavailability of vitamin A were solubility, stability, and dietary composition. Different foods had different bioavailability of vitamin A [42]. However, the recommended dietary reference intake of vitamin A in China for adult men and adult women was 800 ugRAE/day and 700 ugRAE/day, respectively, according to the Chinese dietary reference intakes. The aim of this study was to investigate the amount of vitamin A intake in the Chinese daily diet, rather than the bioavailability of vitamin A. Therefore, effective measures were discussed to increase the vitamin A intake to meet the recommended dietary reference intake of vitamin A.

Firstly, the increase in vegetable consumption is an important measure to obtain the recommended vitamin A dietary intake. Vegetables were rich in vitamins, minerals, and dietary fiber, commonly called the vegetable trio. In addition, the darker vegetables, indicated higher nutritional value [43]. Meanwhile, a survey indicated that vegetables consumption contributes most to dietary happiness over eight days among 14 main food categories [44]. In our study, vegetables had the greatest contributions to vitamin A daily intake among the nine food categories in the Chinese diet, and carrots had the highest content of provitamin A carotenoids and the highest contribution to vitamin A intake among the 16 kinds of vegetables, followed by spinach, Chinese chives, pepper, tomato, asparagus lettuce, lettuce, and pumpkin. Therefore, it was important to improve the amounts of vegetables eaten to increase the vitamin A intake. If one person eats only 1 of the 16 types of vegetables, the amount of each vegetable required is illustrated in Appendix A. If the daily intake of the other 15 vegetables remains unchanged, the amount of each vegetable required to be eaten to meet the RDA of vitamin A intake when coarse cereals, potatoes, legumes, fruits, meat, eggs, milk, and aquatic products are consumed normally is illustrated in Appendix A. When 1 of the 16 types of vegetables was used to replace all of the vegetables, adult women and men required 144 g and 173 g of carrots, 203 g and 244 g of spinach, and 370 g and 445 g of Chinese chives, respectively. It was recommended to intake 300–500 g of vegetables per day according to the Balanced Diet for Chinese Residents; hence, when eating other vegetables, the intake exceeded the recommended daily allowance for vegetables. If the other 15 vegetables remained unchanged in the diet to meet the RDA of vitamin A intake, adult women and men required 100 g and 130 g of carrots, 126 g and 167 g of spinach, 182 g and 257 g of Chinese chives, and 199 g and 276 g of peppers, respectively. The vitamin A daily intake from other vegetables exceeded the recommended daily intake from vegetables. Generally, carrot intake in Western countries was higher than in China due to differences in dietary habits. Therefore, it is hard to adjust the diet of the Chinese population to increase vegetable intake, especially from carrots and spinach, to meet the guideline of vitamin A intake, which was particularly important for patients with eye diseases, neurological disorders, and skin disorders [40].

It could be concluded that vegetables with a high content of provitamin A carotenoids such as carrot and spinach can meet the recommended intake of vegetables. However, it was hard to eat more than 100 g carrots or more than 120 g spinach per day for all Chinese residents. Therefore, improvement of the vitamin A concentration in vegetables was another strategy to increase the vitamin A intake for the entire Chinese population, and this was also the direction for farmers to pursue. The concentrations required for each vegetable to meet the RDA of vitamin A intake when the other 15 vegetables were included in the diet are listed in Table 2. Dark-green or yellow-green vegetables were good for humans, and most of these vegetables were produced in open fields as commodities with lower production costs and prices as well as high yields per unit area. To obtain vitamin A dietary intake of 700 ugRAE/day and 800 ugRAE/day for adult women and men, respectively, the total vitamin A content of spinach should be increased to 529 ugRAE/100 g and 702 ugRAE/100 g, which is 2.18 and 2.89 times higher than the current level, respectively. Similarly, the total vitamin A content of carrots should be increased to 973 ugRAE/100 g and 1256 ugRAE/100 g, which is 2.85 and 3.67 times of the current content, respectively. It was reported that an Indian farmer developed a carrot variety with β-carotene content of 277 mg/kg, which was the recommended amount that our study obtained based on the results [45]. Therefore, it was critical to increase the provitamin A carotenoids’ content of spinach and carrots to meet the daily vitamin A intake. In addition to the common vegetables, vegetables with high provitamin A carotenoids contents were also present in vegetables with low daily consumption. Therefore, we should aim to gradually improve the intake of vegetables with low daily consumption but high provitamin A carotenoids content (e.g., bean curd, amaranth, and kale) and incorporate these into our recipes to increase the intake of vitamin A.

In conclusion, vegetable oil is an ideal matrix to stabilize retinol and delays vitamin A oxidation. Vitamin A can exist in vegetable oil in the form of retinol acetate or retinol palmitate [46]. Therefore, vegetable oil fortification is an important approach to overcome vitamin A deficiency. Rapeseed oil, soybean oil, and peanut oil were the primary types of vegetable oils consumed in China [47], and vitamin A-fortified rapeseed oil and soybean oil were available on the market. These could be used for obtaining the needed vitamin A dietary intake requirements. The recommended intake of vitamin A was 700 ugRAE/day and 800 ugRAE/day for adult women and men, respectively; vegetable oil with 630 ugRAE/100 g and 893 ugRAE/100 g of vitamin A should be implemented into the diets of adult women and men, respectively, when the vegetable oil consumption is 38 g per day. This high value was recommended (893 ugRAE/100 g) as vegetable oil is usually shared by a family. Since lactating women require higher vitamin A intake (1300 ugRAE/day), practical edible oil products with high total vitamin A content for lactating women should be designed. Moreover, the tolerable upper intake level (UL) suggested that the maximum average daily intake may not cause adverse health effects for almost all healthy people. The UL of vitamin A was 3000 ugRAE/day for adults and 700 ugRAE/day for children under 4 years old [38]. Hence, edible oil products with low total vitamin A content for children under 4 years old should be designed. Therefore, the vegetable oil processing industries should design vitamin A-fortified edible oils to meet the recommended daily vitamin A intake.

It has been reported that vitamin A–fortified soybean oil retained 100% of the biological value when the edible oil was heated at 100 °C for 20 min, while it retained 50% of the biological value when the edible oil was used for frying four times at 170 °C [48]. Therefore, vitamin A would not be destroyed during normal cooking. Therefore, consumption of vitamin A-fortified rapeseed oil and soybean oil was a satisfactory way to increase vitamin A intake.

## 5. Conclusions

In summary, vitamin A is widely found in various consumed foods. However, the daily human intake of vitamin A does not currently meet the recommended dietary intake according to scientific and reasonable calculations as well as the vitamin A intake of 460.56 ugRAE/day of Chinese residents. Among the nine different types of foods, vegetables made the greatest contributions to vitamin A intake, accounting for 54.94% of vitamin A intake (253.03 ugRAE/day). In addition, carrots had the highest provitamin A carotenoids’ content, and the contribution of carrots was 28.33% (103.49 ugRAE/day) among the 45 kinds of foods consumed. The current daily intake content is significantly lower than the recommended dietary reference intake of vitamin A (800 ugRAE/day for adult men and 700 ugRAE/day for adult women). Therefore, it is essential to increase vitamin A intake. First, vegetable consumption should be increased to meet the recommended dietary intake. As a result, carrots, spinach, and Chinese chives can meet the recommended daily intake of vegetables. Furthermore, an improvement of the vitamin A concentration in vegetables to meet the recommended dietary intake should be addressed. For example, the concentration in spinach should be increased to 529 ugRAE/100 g and 702 ugRAE/100 g for women and men, respectively. Moreover, increasing the amount of vegetables with low daily consumption but high vitamin A content would be an effective way to increase vitamin A consumption. Additionally, a direct way to increase vitamin A intake is to eat vitamin A-fortified rapeseed oil and soybean oil. Therefore, this study is of great significance to guide the Chinese population to supplement vitamin A intake in a reasonable and scientific manner, thereby meeting the vitamin A dietary intake requirement and reducing the incidence of eye diseases.

## Figures and Tables

**Figure 1 nutrients-15-04028-f001:**
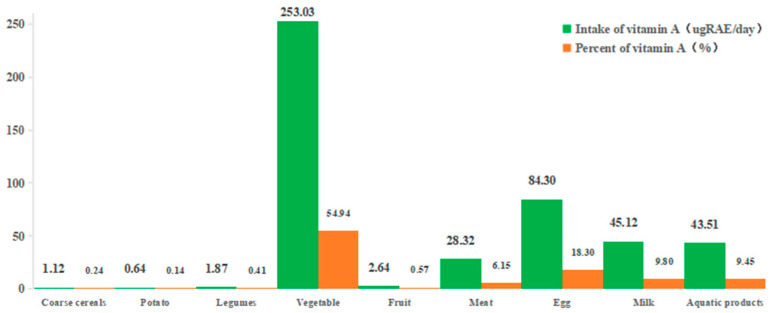
Contribution of different foods to vitamin A dietary intake.

**Table 1 nutrients-15-04028-t001:** Contribution of different kinds of vegetables to vitamin A dietary intake.

Vegetables	Vitamin A Content (ugRAE)-100 g	Consumption (%)	Intake of Vitamin A (ugRAE/day)	Percent of Vitamin A (%)
Carrot	342	11.25	103.48	28.33
Kidney bean	18	2.77	1.34	0.29
Cowpea	10	0.25	0.07	0.02
Eggplant	4	11.40	1.23	0.27
Tomato	31	14.46	12.06	2.62
Red pepper	116	5.43	19.00	4.13
Green pepper	8			
Sweet pepper	6			
Cucumber	8	14.32	3.08	0.67
Pumpkin	74	4.53	9.03	1.96
Chinese chives	133	6.15	22.01	4.78
Cabbage	6	8.57	1.38	0.30
Broccoli	13	4.86	1.70	0.37
Spinach	243	10.22	66.83	14.51
Asparagus lettuce	13	5.78	11.83	2.57
Lettuce	63			
Total			253.03	

**Table 2 nutrients-15-04028-t002:** The increase in vitamin A content in each vegetable needed if the daily intake of the other 15 vegetables remained unchanged.

Vegetables	Area (ha)	Production (hg/ha)	Population (1000 Person)	per Capita Consumption (g/d)	Vitamin A Concentration Required (ugRAE/100 g)/Women/Men	The Current Concentration (ugRAE/100 g)	Multiple (Women/Men)
Carrot	395,525	459,532	141,260	35.251	973/1256	342	2.85/3.67
Kidney bean	876	113,174	141,260	0.019	1,267,271/1,793,586	18	70,403.94/99,643.67
Cowpea	14,391	10,162	141,260	0.028	855,382/1,214,675	10	85,538.2/121,467.5
Eggplant	804,381	465,690	141,260	72.652	331/469	4	82.75/117.25
Tomato	1,144,821	590,806	141,260	131.181	192/268	31	6.19/8.65
Red pepper	754,718	221,933	141,260	32.486	796/1103	116	6.12/8.48
Green pepper						8	
Sweet pepper						6	
Cucumber	1,292,545	584,874	141,260	146.621	165/234	8	20.63/29.25
Pumpkin	401,581	185,266	141,260	14.430	1722/2415	74	23.27/32.64
Chinese chives	5625	251,302	141,260	0.274	95,419/131,915	133	717.44/991.84
Cabbage	1,002,516	350,046	141,260	68.062	354/501	6	59/83.5
Broccoli	484,031	198,477	141,260	18.633	1294/1831	13	99.54/140.85
Spinach	714,572	417,691	141,260	57.888	529/702	243	2.18/2.89
Asparagus lettuce	607,979	236,321	141,260	27.866	902/1260	13	11.87/16.58
Lettuce						63	

## Data Availability

The data that support the findings of this study are available on request to the corresponding author.

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
