# Peer review of "Contribution of Different Food Types to Vitamin A Intake in the Chinese Diet"

_nutrients, 2023, doi:10.3390/nu15184028_

Round 1

Reviewer 1 Report

Line 30: Please, include examples of provitamin A carotenoids.

Line 45: What is the percentage of VAD incidence in each sex?

Line 47: 1020%? Wouldn’t it be 10.20%? This needs to be checked. Same issue on line 59.

Line 136: 1088 ug RAE/100 g, right? The “/100g” needs to be added. Here and elsewhere.

Section 3.2: It is not clear to me how the consumption (g/day) has been estimated. Those quantities were based on what kind of data?

Lines 179-180: The actual data needs to be cited along with appropriate references. Saying “a large proportion” is not enough to back up the statement. 

Line 212: “It can be concluded” instead of “it can conclude.”

Lines 212-214: Sentence needs to be rewritten.

Line 218: “listed”, not “list.”

Comment: I believe this manuscript has a major flaw: there is no discussion about the bioavailability aspect of vitamin A in these different sources. Some sources do provide the nutrient, but the bioaccessibility and bioavailability are so low that there is no point in increasing their consumption. This is highly variable according to the food matrix and should definitely be a point of discussion in an article like this. Therefore, I would like to see this subject addressed in the discussion section with a certain level of depth.

I have noticed spelling errors, grammar mistakes, and unusual sentence constructions throughout the manuscript. Therefore, this aspect needs to be revised before resubmission.

Author Response

Reviewer #1: Comments and Suggestions for Authors

Line 30: Please, include examples of provitamin A carotenoids.

Response: following the suggestion from the reviewer, example of provitamin A carotenoids was added into the revised manuscript.

‘One is preformed vitamin A(retinol), and the other is provitamin A carotenoids comprised chiefly of beta-carotene, alpha-carotene and cryptoxanthin [1-2]’.

Line 45: What is the percentage of VAD incidence in each sex?

Response: following the suggestion from the reviewer, the percentage of VAD incidence in each sex was added into the revised manuscript.

‘Children are commonly found to have VAD, especially preschool-age children, and boys and girls possess the incidence of 0.8% and 0.4%, respectively [15-17]’.

Line 47: 1020%? Wouldn’t it be 10.20%? This needs to be checked. Same issue on line 59.

Response: 10-20% and 20-34% are correct. We have revised them in lines 47 and 59.

Line 136: 1088 ug RAE/100 g, right? The “/100g” needs to be added. Here and elsewhere.

Response: following the suggestion from the reviewer, we have revised it in the revised manuscript.

Section 3.2: It is not clear to me how the consumption (g/day) has been estimated. Those quantities were based on what kind of data?

Response: In the section of Materials and methods, data sources described the sources of vitamin A content and consumption of various foods and calculation method introduced the specific calculation process.

‘ 2.1. Data sources

Cereals, vegetable oils, and nuts do not contain vitamin A. Therefore, coarse cereals, potatoes, legumes, vegetables, fruits, meat, eggs, milk, and aquatic products are used to obtain vitamin A according to the Chinese Food Composition Tables released by National Institute for Nutrition and Health and Chinese Center for Disease Control and Presentation. Production, supply, and distribution (PSD) reports released by the United States Department of Agriculture (USDA), the Food and Agriculture Organization of the United States (FAOSTAT), China Population Nutrition and Health Status Monitoring Report and the China Statistical Yearbook provide the domestic consumption of main kinds of foods in China. Cereals, coarse cereals, potatoes, legumes, vegetables, fruits, nuts, vegetable oils, meat, eggs, milk, and aquatic products as the consumption of major foods in Chinese diet are concluded from the China Population Nutrition and Health Status Monitoring Report and USDA PSD reports. The most and frequent consumed 16 kinds of vegetables are selected and the consumption of each kind of vegetable are obtained from FAOSTAT while the USDA PSD reports provide the major fruits, which includes apple, banana, pear, grape, peach, orange, grapefruit, and cherry. In summary, 45 kinds of foods are used to evaluate the vitamin A intake in the Chinese diet.

2.2. Calculation Method

The United States Health and Medicine Division (HMD) defined the total vitamin A was the sum of 1 μg retinol, 1/12 μg of dietary β-carotene, and 1/24 μg of dietary α-carotene, which was expressed as μg of retinol activity equivalents (RAE). First, the estimated daily vitamin A intake was calculated by total vitamin A content and the consumption of various foods that are common consumed, meanwhile, the contribution of one kind food to total vitamin A intake was calculated. Second, the amount of each vegetable required to eat when a person only ate one of the 16 types of vegetables was calculated. Similarly, the amount of each vegetable required to eat when the other 15 vegetables in daily intake remained unchanged was calculated. In addition, the vitamin A content in each vegetable that needed to be increased when the daily intake of the other 15 vegetables remained unchanged was calculated’.

Lines 179-180: The actual data needs to be cited along with appropriate references. Saying “a large proportion” is not enough to back up the statement. 

Response: following the suggestion from the reviewer, we have revised it in the revised manuscript as follows:

Nearly 254 million of preschool-aged children were affected by VAD and suffered from night blindness [40]’.

Line 212: “It can be concluded” instead of “it can conclude.”

Response: following the suggestion from the reviewer, we have revised it in the revised manuscript.

Lines 212-214: Sentence needs to be rewritten.

Response: following the suggestion from the reviewer, we have revised it in the revised manuscript.

‘It can be concluded that vegetables with high content of provitamin A carotenoids  such as carrot and spinach can meet the recommended intake of vegetables. However, it is hard to eat more than 100 g carrots or more than 120 g spinach per day for all Chinese residents’.

Line 218: “listed”, not “list.”

Response: following the suggestion from the reviewer, we have revised it in the revised manuscript.

Comment: I believe this manuscript has a major flaw: there is no discussion about the bioavailability aspect of vitamin A in these different sources. Some sources do provide the nutrient, but the bioaccessibility and bioavailability are so low that there is no point in increasing their consumption. This is highly variable according to the food matrix and should definitely be a point of discussion in an article like this. Therefore, I would like to see this subject addressed in the discussion section with a certain level of depth.

Response: following the suggestion from the reviewer, we have revised it in the revised manuscript.

‘Bioavailability was defined that the fraction of ingested bioactive agent reaches the specific site of action in the body determines its efficacy, which primarily determined by three factors: bioaccessibility, transformation and absorption [47]. The main factors limiting the bioavailability of vitamin A are solubility, stability and dietary composition, different food have different bioavailability of vitamin A [48]. However, the recommended dietary reference intake of vitamin A for adult men is 800 ugRAE/day and for adult women is 700 ugRAE/day in China according to the Chinese dietary reference intakes. The aim of this study is about to meet the recommended dietary reference intake of vitamin A, not to consider the bioavailability of vitamin A. Therefore, measures above mentioned are effective to increase the vitamin A intake to meet the recommended dietary reference intake of vitamin A’.

Comments on the Quality of English Language

I have noticed spelling errors, grammar mistakes, and unusual sentence constructions throughout the manuscript. Therefore, this aspect needs to be revised before resubmission.

Response: following the suggestion from the reviewer, we have carefully revised the article.

Reviewer 2 Report

In the manuscript submitted to me for review entitled "Contribution of different food types to vitamin A intake in the Chinese diet", the authors Xue Li, Can Guo, Yu Zhang, Li Yu, Fei Ma, Xuefang Wang and Liangxiao Zhang present the content of vitamin A in various vegetables included in the daily diet of the population in China.

The study has an important scientific significance because in recent years an increasing number of people suffering from obesity has been observed. This leads to different diets, which lead to a lack of important nutritional elements, including vitamin A. As a result, a number of health problems can occur. Adequate intake of vital nutrients is of utmost importance and the present manuscript contributes to this by examining the content and, accordingly, daily requirements of vitamin A intake.

The authors' study covers scientific data published in the last three decades. To support their research, the authors use 45 references, of which 21 (almost 1/2 of the total number) are from the last 5 years.

My remarks and recommendations to the authors are:

1. On line 47, a percentage value of 1020% is presented. I guess a decimal point was dropped, but let the authors check the value anyway and correct it. Same note for line 59 - it has a value of 2034%.

2. On line 112, the word "orange" is repeated - delete one copy.

3. In the "Methods" section, the actual methodology for determining the amounts of vitamin A is not given. Nor by which methodology were the presented values determined, nor what consumables and equipment was used for this. It would be good if the authors describe how the presented values for the amount of vitamin A in food products were determined. In the version thus presented, the methodology could not be repeated by another research team, because it was not presented at all.

Author Response

Reviewer #2: Comments and Suggestions for Authors

In the manuscript submitted to me for review entitled "Contribution of different food types to vitamin A intake in the Chinese diet", the authors Xue Li, Can Guo, Yu Zhang, Li Yu, Fei Ma, Xuefang Wang and Liangxiao Zhang present the content of vitamin A in various vegetables included in the daily diet of the population in China.

The study has an important scientific significance because in recent years an increasing number of people suffering from obesity has been observed. This leads to different diets, which lead to a lack of important nutritional elements, including vitamin A. As a result, a number of health problems can occur. Adequate intake of vital nutrients is of utmost importance and the present manuscript contributes to this by examining the content and, accordingly, daily requirements of vitamin A intake.

The authors' study covers scientific data published in the last three decades. To support their research, the authors use 45 references, of which 21 (almost 1/2 of the total number) are from the last 5 years.

My remarks and recommendations to the authors are:

  1. On line 47, a percentage value of 1020% is presented. I guess a decimal point was dropped, but let the authors check the value anyway and correct it. Same note for line 59 - it has a value of 2034%.

Response: 10-20% and 20-34% are correct. We have revised them in lines 47 and 59.

  1. On line 112, the word "orange" is repeated - delete one copy.

Response: following the suggestion from the reviewer, we have revised it in the revised manuscript.

  1. In the "Methods" section, the actual methodology for determining the amounts of vitamin A is not given. Nor by which methodology were the presented values determined, nor what consumables and equipment was used for this. It would be good if the authors describe how the presented values for the amount of vitamin A in food products were determined. In the version thus presented, the methodology could not be repeated by another research team, because it was not presented at all.

Response: we did not detect the vitamin A in foods but use the data in the Chinese Food Composition Tables. In the section of ‘Data sources’, data collection of vitamin A content was described in detail.

‘Cereals, vegetable oils, and nuts do not contain vitamin A. Therefore, coarse cereals, potatoes, legumes, vegetables, fruits, meat, eggs, milk, and aquatic products are used to obtain vitamin A according to the Chinese Food Composition Tables released by National Institute for Nutrition and Health and Chinese Center for Disease Control and Presentation’.

Round 2

Reviewer 1 Report

The manuscript has been improved as the authors incorporated all the feedback given in the previous revision. Therefore, I suggest it be published in its current form.

Minor errors are still detected.